# A classification framework for identifying bronchitis and pneumonia in children based on a small-scale cough sounds dataset

Siqi Liao[1], Chao Song[1], Xiaoqin Wang[2]☯*, Yanyun Wang[3]☯*

**1** School of Computer Science and Engineering, University of Electronic Science and Technology of China, Chengdu, Sichuan, China, **2** Department of Pediatric Cardiology, Key Laboratory of Birth Defects and Related Diseases of Women and Children (Sichuan University), Ministry of Education, West China Second University Hospital, Sichuan University, Chengdu, Sichuan, China, **3** Laboratory of Molecular Translational Medicine, Center for Translational Medicine, Key Laboratory of Birth Defects and Related Diseases of Women and Children (Sichuan University), Ministry of Education, West China Second University Hospital, Sichuan University, Chengdu, Sichuan, China

☯ These authors contributed equally to this work.
* 996511844@qq.com (XW); wangyanyun@scu.edu.cn (YW)

**Data Availability Statement:** All the data that support the findings of this study are available from (https://doi.org/10.6084/m9.figshare.21176197.v1).

## Abstract

Bronchitis and pneumonia are the common respiratory diseases, of which pneumonia is the leading cause of mortality in pediatric patients worldwide and impose intense pressure on health care systems. This study aims to classify bronchitis and pneumonia in children by analyzing cough sounds. We propose a Classification Framework based on Cough Sounds (CFCS) to identify bronchitis and pneumonia in children. Our dataset includes cough sounds from 173 outpatients at the West China Second University Hospital, Sichuan University, Chengdu, China. We adopt aggregation operation to obtain patients' disease features because some cough chunks carry the disease information while others do not. In the stage of classification in our framework, we adopt Support Vector Machine (SVM) to classify the diseases due to the small scale of our dataset. Furthermore, we apply data augmentation to our dataset to enlarge the number of samples and then adopt Long Short-Term Memory Network (LSTM) to classify. After 45 random tests on RAW dataset, SVM achieves the best classification accuracy of 86.04% and standard deviation of 4.7%. The precision of bronchitis and pneumonia is 93.75% and 87.5%, and their recall is 88.24% and 93.33%. The AUC of SVM and LSTM classification models on the dataset with pitch-shifting data augmentation reach 0.92 and 0.93, respectively. Extensive experimental results show that CFCS can effectively classify children into bronchitis and pneumonia.

## Introduction

Each year, childhood respiratory infections significantly burden our healthcare system regarding staffing and resource utilization [1,2]. Statistics from one of the largest children's hospitals in China showed that bronchitis and pneumonia are the leading respiratory diseases in outpatient during the decade [3], of which pneumonia has been the leading cause of death under 5

**Funding:** This work is supported by the National Key R&D Program of China under Grant 2021YFB3101302 (Recipient: Chao Song) and 2021YFB3101303(Recipient: Chao Song); the National Natural Science Foundation of China under Grant No.62020106013(Recipient: Chao Song); the Technology Achievements Transformation Demonstration Project of Sichuan Province of China No.2018CC0094(Recipient: Chao Song); and the Fundamental Research Funds for the Central Universities No. ZYGX2019J075(Recipient: Chao Song), 2082604401036(Recipient: Yanyun Wang).

**Competing interests:** The authors have declared that no competing interests exist.

years old worldwide for the past three decades. It is estimated that 1.2 million children under the age of five die of pneumonia yearly [4]. COVID-19 is one kind of pneumonia, but we only discuss ordinary pneumonia in this paper. Bronchitis is the primary infection of the lower respiratory tract in children under two [5–7]. Pneumonia and bronchitis present highly similar symptoms, such as persistent cough, fever, and rapid breathing rate. Highly similar symptoms make it difficult to diagnose the two diseases [8].

Furthermore, the number of outpatients is increasing, which has led to overcrowding in the outpatient clinic [9]. Numerous computer-aided diagnostic systems have been developed to help diagnose diseases [10,11]. Doctors can use the results of the computer-aided system as a reference, saving patients' waiting time and reducing doctors' misdiagnosis rate.

The goal of this paper is to classify patients with bronchitis and pneumonia. From the view of the pathology, cough sounds are generated from the lungs and respiratory tract, carrying information about abnormalities in the lungs and respiratory tract. By analyzing cough sounds, we can learn the types and severity of respiratory diseases [12]. Moreover, pneumonia and bronchial patients' cough are more easily observed than other symptoms. So we use cough sounds to classify bronchitis and pneumonia.

The process of cough sound recognition is similar to speech recognition [13]. Firstly, pre-process the cough signals before extracting the features of the diseases, and then the extracted features are input into the trained classification model to obtain the classification results.

We term the recorded audios of patients (children) with respiratory problems during clinic visits as patient audios. As the patient audios are collected in the hospital, noise and human speaking voices are inevitably in them, so we cannot directly extract the disease feature from the patient audios. Therefore, we segment them into several chunks based on the energy threshold. We term those chunks only containing the cough sounds signal as cough chunks. This study will extract features from the cough chunks as input to the model.

However, designing such a cough sounds classification model involves two complex challenges. (1) Disease labels do not correspond to the cough chunks' features. It is because some cough chunks carry the disease information while others do not. However, what we have are patient audios and patient disease labels. (2) The dataset collected from the hospital is small. Patients are concerned about their privacy. It is difficult to collect large amounts of cough sound data because it is a kind of private data. The small scale of a dataset may lead to a low classification accuracy of the diseases.

In this paper, we propose a Classification Framework based on Cough Sounds (CFCS) for identifying bronchitis and pneumonia in children. Overview of the CFCS as shown in Fig 1. It segments patient audios into multiple cough chunks because they contain unrelated speech voices. And then, it aggregates several cough chunks' features. The aggregated results, which represent the feature of the patient audio, are used as the model's input to classify. In summary, our main contributions are as follows:

1. We proposed a feature aggregation operation to obtain patients' disease features to classify the diseases.

2. We proposed two strategies to solve the problem of the small scale of a medical dataset. One is to use SVM on the small-scale dataset; the other is to adopt data augmentation before using LSTM classification.

3. We have collected a cough sounds dataset, and our comparative experiments on the real-world dataset show that CFCS can achieve a good classification effect.

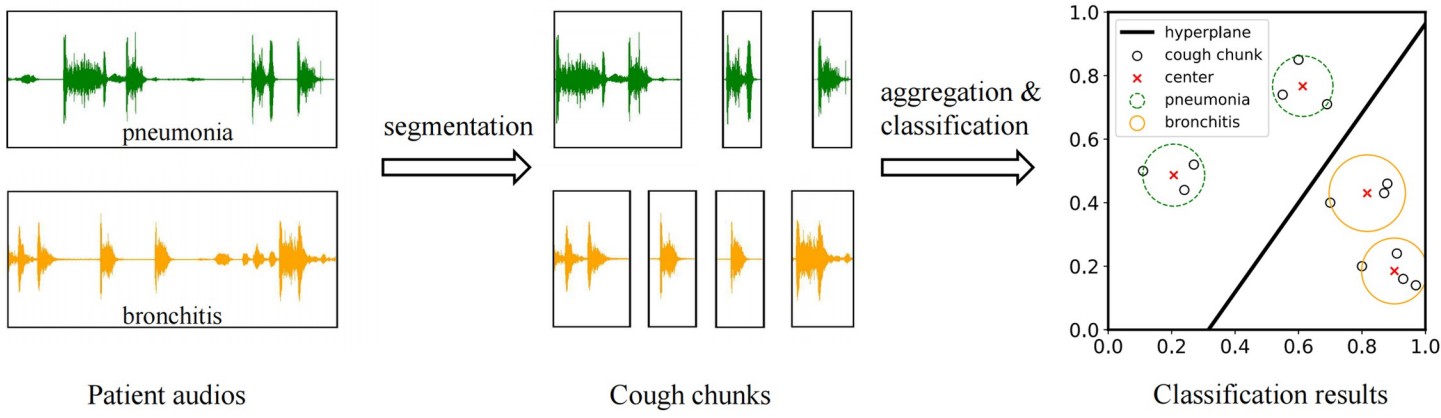

**Fig 1. Overview of the Classification Framework based on Cough Sounds (CFCS).**

## Methods

### Overview of construction of classification models

We first state the process of construction of classification models for diagnosing patients with bronchitis and pneumonia, then introduce the details of collecting patient audios, and finally show the statistics of the patient audios. Table 1 shows the notations used in this paper.

### The process of construction of classification models

We use the training set to learn a cough sound classification model and then use the model to predict the test data. We formalize the problem of disease classification. Let $T = \{(x_1, y_1), (x_2, y_2), \cdots, (x_N, y_N)\}$ denote the training set, where $(x_i, y_i)$ is a sample, $i = 1, 2, \cdots, N$. Let $x_i$ denote the features that the model used for classification. Let $y_i$ denote the diagnostic result of $x_i$, i.e., $y_i = \{-1, +1\}$. $y_i = -1$ if $x_i$ is the negative case (bronchitis) and $y_i = +1$ otherwise.

In the learning process, the learning system uses the training set $T$ to learn a classification decision function. The classification decision function is denoted by $Y = f(X)$, which describes the mapping between inputs and outputs.

In the prediction process, for an input $x_{N+1}$ in the given test set, the prediction system outputs a classification result $\hat{y}_{N+1}$ by the model.

**Table 1. Notations.**

| Notation | Interpretation |
|---|---|
| $T$ | Training set |
| $N$ | Number of the training sample |
| $(x_i, y_i)$ | A sample (feature, label) |
| $Y = f(X)$ | Mapping between features and labels |
| $x_{ki}$ | Feature matrix of the $i^{th}$ cough chunk of the patient $k$ |
| $x_{ki}^{(t)}$ | The $i^{th}$ feature vector of $x_{ki}$ |
| $x_k$ | Feature matrix of the patient $k$ |
| $w, b$ | Normal vector, intercept of the hyperplane |
| $\zeta, C$ | Relaxation variable, regularization parameter |

## Collection of patient audio

The patient audios used in this work are collected from the West China Second University Hospital, Sichuan University, Chengdu, China. The Ethics Committees of West China Second University Hospital approved the study and the verbal consent procedures. Verbal informed consent was obtained from the legal guardians of all participants and recorded with the recorder. We collect 173 audios from 91 bronchitis (51 male, 40 female; 1 acute asthmatic bronchitis, 13 acute bronchiolitis,76 acute bronchitis) and 82 pneumonia patients (43 male, 39 female; 1 lobar pneumonia, 81 bronchopneumonia) (ages from 0 years to 11 years). Bronchitis and pneumonia were diagnosed according to Zhu Futang Practice of Pediatrics (8th Edition) [14]. Fig 2 shows the age distributions of bronchitis, pneumonia, and all patients. The proportion of age represents the ratio of the patients in each age group to the number of patients. As shown in Fig 3, patient audios are collected in a consulting room of pediatrics as MP3 files. The distance between the recorder and the patient's mouth varies from 20 to 40 cm. The average duration of the audio for each patient is 3.92s. In addition, children are usually accompanied by their families, who can make some extra noise for the audios.

## Statistics of the dataset

Table 2 shows the detailed statistics of the patient audios, and each disease accounts for about half of the dataset. Intuitively, the statistical data approximately follows a power-law distribution, and the proportion of children under the age of one accounts for more than 50%.

## Structure of the feature aggregation framework

The core of the framework is aggregation operation, which can obtain the patient's features. Fig 4 shows the structure of the feature aggregation framework. We first take a recording of patients, followed by noise reduction and normalization. Then, we segment the patient audios into several cough chunks. In addition, we apply three data augmentation technologies to the cough chunks. Later, we extract MFCC features from the cough chunks. Finally, we train a classifier to classify pneumonia and bronchitis.

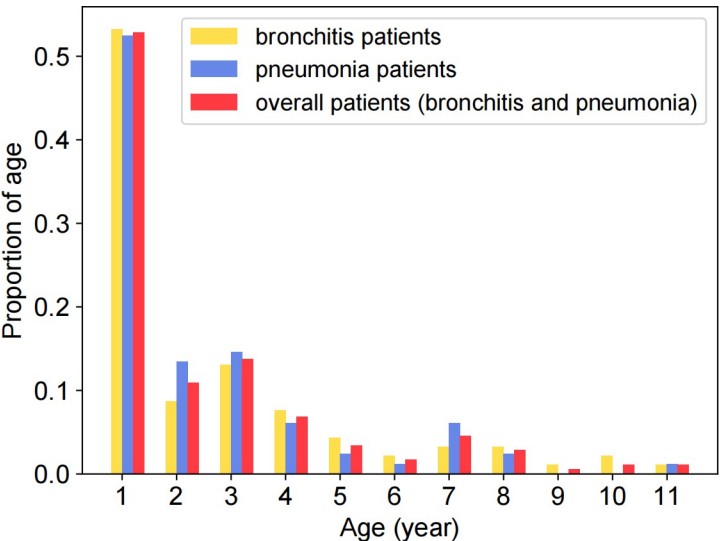

**Fig 2. Age distributions of bronchitis, pneumonia, and all patients.**

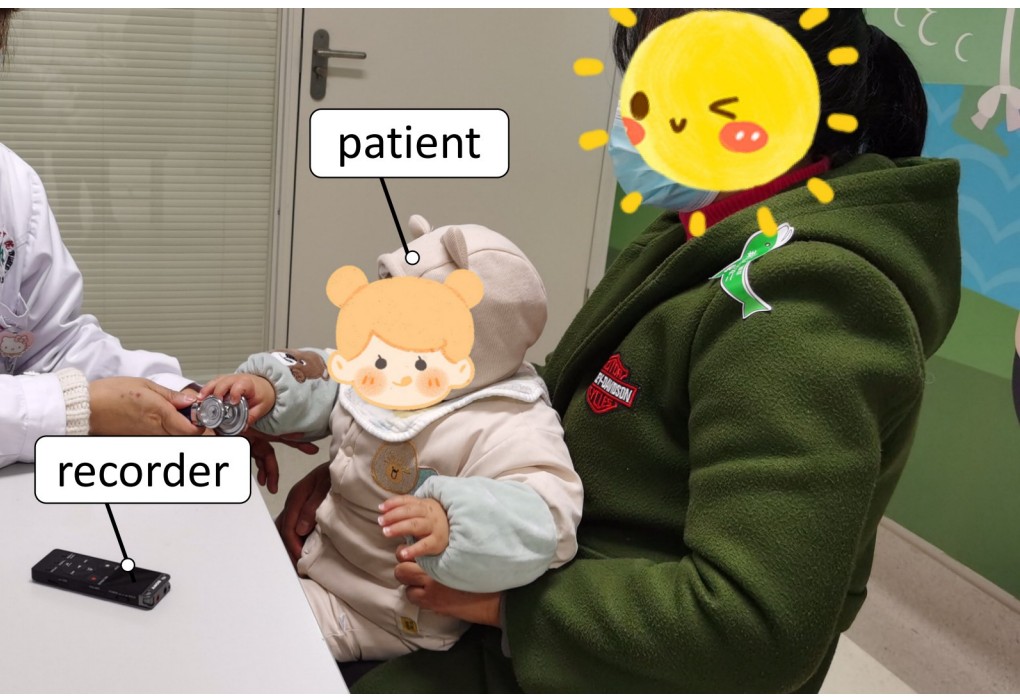

**Fig 3. Recording setup in the hospital.** We collected patient audios in a pediatrics consulting room of the West China Second University Hospital, Sichuan University, Chengdu, China.

**Data pre-processing.** We will start by improving the signal-to-noise ratio (SNR) of the patient audios. We first convert patient audios into WAV format at 44.1kHz sampling frequency with 16 bits per sample. Fig 5A shows the waveform of the original patient audio. We adopt Log-MMSE [15], a frequently used speech enhancement algorithm [16], to improve the SNR. It minimizes the mean square error of the log-spectral, resulting in a much lower residual noise level without further affecting the patient audio itself. In addition, we normalize the amplitude value of the patient audios by limiting the peak amplitude to -0.1dB. Fig 5B shows the waveform after speech enhancement and normalization.

**Patient audios segmentation.** After data pre-processing, patient audios still contain people's speaking voices. So we further need to segment patient audios into cough chunks. There are many audio segmentation algorithms. A widely adopted algorithm for audio segmentation is based on the Bayesian Information Criterion (BIC), applied within a sliding variable-size analysis window and some smoothing rules. The sliding variable-size analysis window can classify each one-second window into different audio classes by audio signals features. The smoothing rules of an audio sequence can segment an audio stream into speech, music, environment sound and silence [17,18].

Auditok is fast and works well for audio streams with low background noise (e.g., few people talking) [19]. Auditok uses a signal energy threshold to obtain valid audio events, where the valid audio events are those signal energy equal to or above this threshold. Moreover, the energy of the audio signal is the log energy, which is computed as: $20lg\sqrt{\frac{1}{N} \cdot \sum_{i=1}^{N} a_i^2}$, where $a_i$ is the $i^{th}$ audio sample and $N$ is the number of audio samples in data. Meanwhile, our dataset is collected in a quiet environment with few people, so we use a toolkit called auditok to segment the patient audios.

**Table 2. Detailed statistics of patient audios collected in the hospital.**

| # | last(s) | class | # | percentage(%) | last(s) | category | # | percentage(%) | last(s) |
|---|---|---|---|---|---|---|---|---|---|
| 173 | 3.92 | Bronchitis | 91 | 52.6 | 3.4 | Bronchitis | 80 | 46.2 | 2.37 |
| | | | | | | Asthmatic bronchitis | 5 | 2.9 | 4.85 |
| | | | | | | Bronchiolitis | 6 | 3.5 | 15.96 |
| | | Pneumonia | 82 | 47.4 | 4.49 | Pneumonia | 55 | 31.8 | 2.94 |
| | | | | | | Bronchopneumonia | 23 | 13.3 | 7.71 |
| | | | | | | Lobular pneumonia | 4 | 2.3 | 7.22 |

This table shows total audio statistics, statistics for the two categories, and more granular disease category statistics for the two categories. Note that the "#" column shows the number of each category, the "percentage" column reflects the percentage of the total for each category, and the "last" column shows the average duration of each category of patient audios collected.

In the data collection process, the distance between the recorder and the patient's mouth is no more than 40cm. Statistical analysis shows us an energy threshold value to discard speaking voice from cough sounds. Therefore, we used this energy threshold to segment cough sounds. Fig 5C shows the waveform after patient audios segmentation according to the threshold. Cough chunks are retained and speaking voices are discarded. In the future, we can use a sliding variable-size analysis window to perform segmentation in the scene of a complex environment.

**Data augmentation.** As medical data is a kind of private data, it is expensive to collect such private data. Furthermore, deep learning relies on a large-scale dataset. Data augmentation aims to increase the number and diversity of training data to improve the robustness of

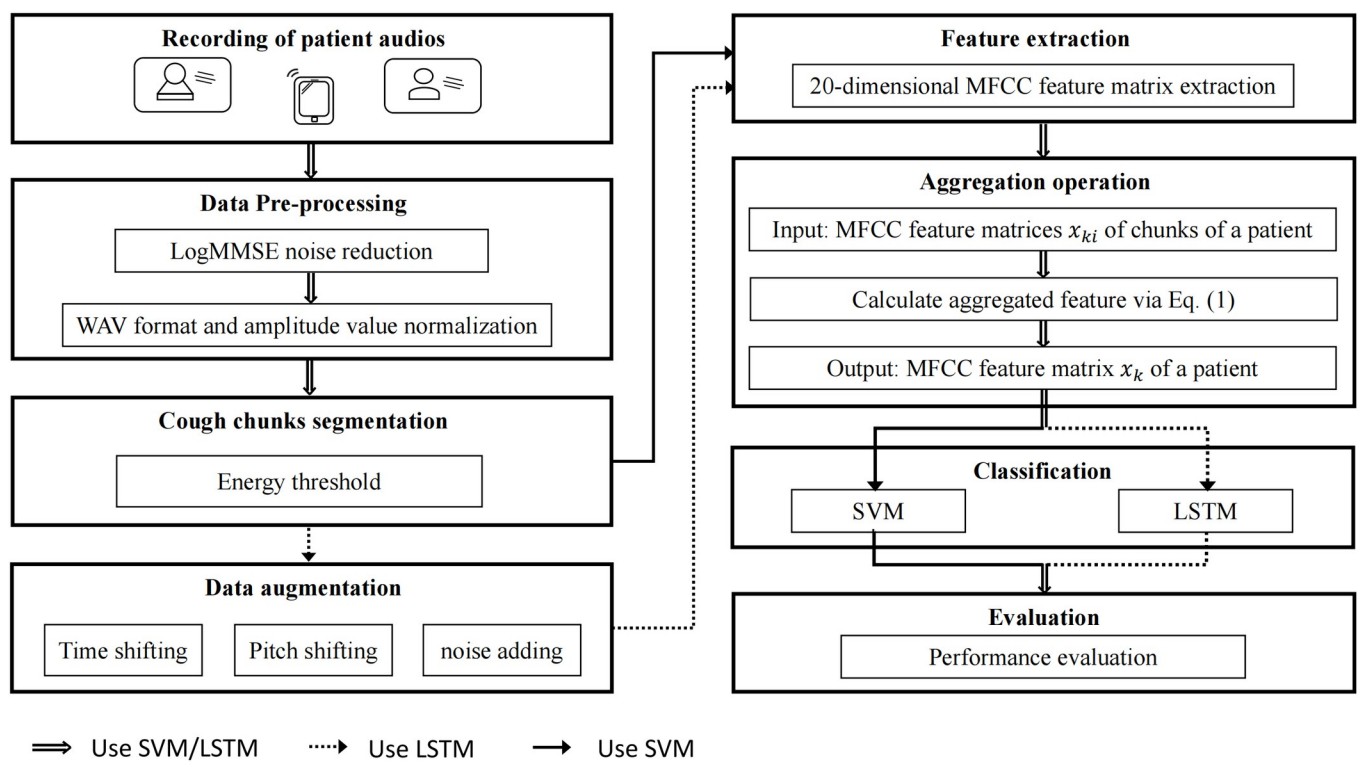

**Fig 4. Structure of CFCS for classifying bronchitis and pneumonia in children.**

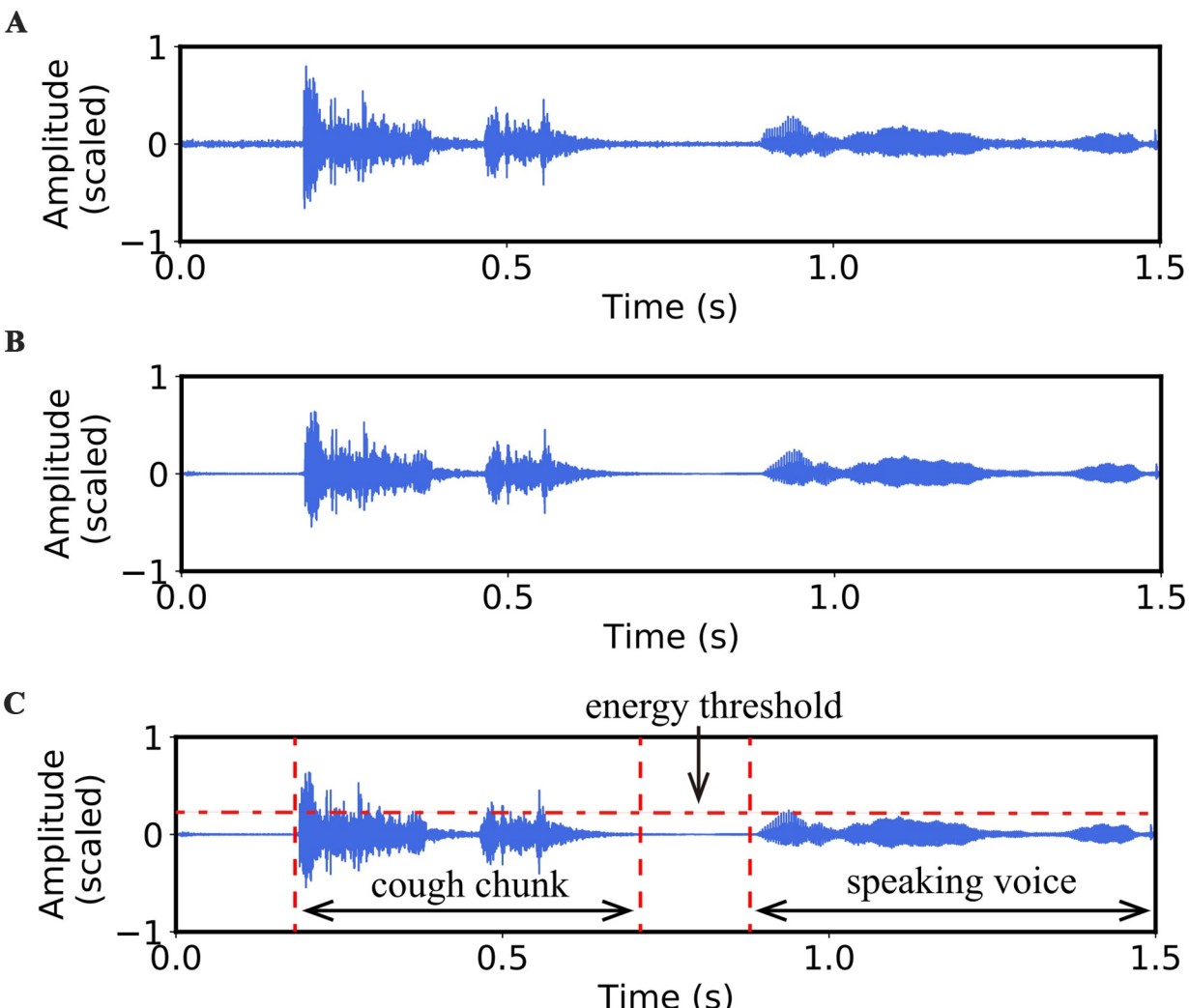

**Fig 5. Example of preprocessing and patient audio segmentation waveforms.** (A) Waveform of the original patient audio; (B) Waveform after noise reduction and normalization; (C) Waveform after patient audios segmentation according to the energy threshold.

deep learning models. We term the cough sounds collected in the hospital as the RAW dataset. We have adopted three data augmentation technologies: time shifting [20], pitch shifting [21] and adding noise [22] to the RAW dataset. Fig 6 shows the waveforms of the cough chunks through the three data augmentation methods, where the cough chunks are derived from the above selection.

*Time shifting*. We adopt time shifting to increase the number of samples of the RAW dataset. This operation can be seen as deleting a small portion of cough sound information to obtain new samples. Time shifting deletes the information at the beginning or end of the cough chunks, where the period ranges from 0 to 0.1s, and fills fixed frequency to keep the duration unchanged.

*Pitch shifting*. We also adopt pitch shifting to increase the number of samples of the RAW dataset. Raise the pitch of the cough chunks within five half-tones. That is, turn up the frequency. The higher the pitch, the higher the frequency. We can obtain new cough sounds through raw cough chunks by conducting pitch shifting.

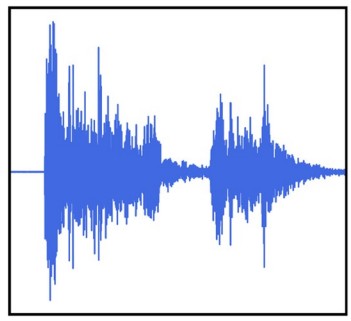 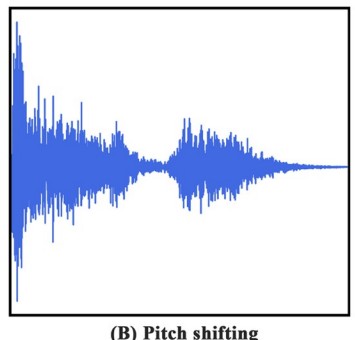 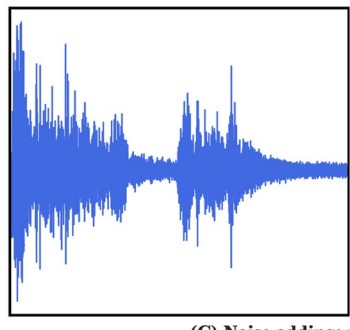 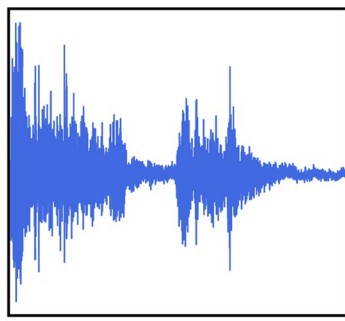

| (A) Time shifting | (B) Pitch shifting | (C) Noise adding: white (left) / pink (right) noise |

**Fig 6. Waveforms of the cough chunks under three data augmentation methods.**

*Noise adding (white/pink noise).* To increase the diversity of samples of the RAW dataset, we mix noise with the original sounds. This operation can be seen as changing the SNR distribution of each cough chunk. Mix each cough chunk with white noise or pink noise. White noise contains various characteristics of noise. Pink noise is the most common noise in nature, and traffic sound can be simulated by pink noise. So we mix the white and pink noise to the cough chunks to obtain new cough sounds.

**Feature extraction.** In speech recognition, there are many feature extraction methods [23–25]. We extracted the Mel frequency cepstral coefficients (MFCC) [26] from the cough chunks using a non-parametric FFT-based approach. MFCC describes the energy distribution of a signal in the frequency domain. The dimension of MFCC depends on the front part number of dimensions taken from the data after discrete cosine transform (DCT). Because a lot of the signal data will be transformed in the low-frequency region after DCT, it is only necessary to take the front part after DCT and discard the redundant data. MFCC is frequently used as an acoustic feature to assess pathological voice quality [27,28]. This study uses a 20-dimensional feature vector consisting of log energy and 19-dimensional Mel frequency cepstral coefficients.

The process of computing MFCC involves four steps: (1) Divide the audio stream into overlapping frames. (2) Perform an FFT for each frame to obtain the frequency spectrum. (3) Then, take the logarithm on the spectrum and convert the log spectrum to the Mel spectrum. (4) Finally, take the Discrete Cosine Transform (DCT) on the Mel spectrum

**Aggregation operation.** Some cough chunks carry information about the disease, while others are not. In reality, we do not know the label of cough chunks in the dataset. We only have the patient audios and disease labels. Thus we want to analyze all the cough chunks comprehensively.

We first intuitively concatenate the cough chunks on the time sequence. E.g., starting from 0s, if the first cough chunk lasts $t_1$s, then the second cough chunk starts from $t_1$s, following the first cough chunk, and so on. We obtained the experimental results of SVM [29], XGBoost [30], RF [31], LSTM [32], RNN [33], and GRU [34] classification accuracy in 45 random test on the RAW dataset, 69.79%, 64.38%, 67.5%, 66.46%, 62.5%, and 66.25%, respectively. From the above results, there seems to be much room for improvement in classification accuracy. So, we further take into account the interconnection of each cough chunk.

As we all know, the binary classification result is determined by the distribution of features. For example, as shown in the third part of Fig 1, the features of cough chunks are all distributed in the same feature space, and the hyperplane divides these features into two classes. If multiple cough chunks of one patient audio are mapped in the same class and are far from the

hyperplane, the patient is more likely to belong to this class. Furthermore, the mean value of all cough chunks (i.e., the center value in Fig 1) can comprehensively represent all cough chunks.

Then, we utilize the mean value of the features of all cough chunks in one patient audio as the patient's disease feature. Formally, the feature extracted from the $i^{th}$ cough chunk of the patient $k$ can be defined as $x_{ki} = \left( x_{ki}^{(1)}, x_{ki}^{(2)}, \cdots, x_{ki}^{(20)} \right)^T$, which is a 20-dimensional feature vector, where $x_{ki}^{(t)}$ is the $t^{th}$ feature of the $x_{ki}$. From the above, we set the feature of the patient $k$ as follows:

$$x_k = \frac{1}{n} \left( \sum_{i=1}^{n} x_{ki}^{(1)}, \sum_{i=1}^{n} x_{ki}^{(2)}, \cdots, \sum_{i=1}^{n} x_{ki}^{(20)} \right)^T$$

**Classification.** Support vector machine (SVM) [29] is a powerful data mining technique for classifying data. SVM classifies data by constructing a linear or non-linear separating hyperplane from the training set. SVM separates the two classes while maximizing the margin between this hyperplane and the two classes. When the data is linearly in-separated in low dimensions, it needs to transform the original data into a higher-dimensional space. SVM can use kernel function, a mathematical trick that avoids the overhead of data computation in high dimensions. Moreover, there are three frequently used kernel functions: linear kernel function, Gaussian kernel function, and radial basis function.

In general, training sets may still not be linearly separable in feature space. SVM adopts a relaxation variable to solve this problem. Mathematically, the optimization problem is formulated as:

$$\min_{w,b,\xi} \frac{1}{2} \|w\|^2 + C \sum_{i=1}^{N} \xi_i$$
$$s.t. \ y_i(w \cdot x_i) + b \geq 1 - \xi_i, i = 1, 2, \cdots, N$$
$$\xi_i \geq 0, i = 1, 2, \cdots, N$$

where $w$ and $b$ denote normal vector and intercept of the separating hyperplane, respectively, $\zeta$ denotes relaxation variable, $C$ denotes regularization parameter, and $N$ denotes the number of samples. Consider relaxation variables as costs, which can be adjusted by regularization parameter $C$. The higher the value of $C$, the greater the penalty for misclassification; otherwise, the smaller the penalty.

We expect to make the classification margin as large as possible and the number of misclassified samples as little as possible. We can calculate an optimal solution $w^*$ and $b^*$ for the objective function, and obtain the separating hyperplane: $w^* \cdot x + b^* = 0$. And the classification decision function can be represented by $f(x) = sign(w^* \cdot x + b^*)$.

Long Short-Term Memory Network (LSTM) [32] is a recurrent neural network variant that can effectively process sequence information. As shown in Fig 7, we adopt this LSTM sequence model to classify the diseases. The core of LSTM is to adopt the gate structure, including input gate, output gate and forget gate (i.e., retain the important information and discard the unimportant information), so it can effectively solve the problems of the gradient disappearance and the long-term dependence.

Given an input sequence $X = \{x_1, x_2, \cdots, x_N\}$, where $x_k$ is a 20-dimensional feature vector, $k = 1, 2, \cdots, N$. The batch size refers to the number of samples input during training. The time step refers to the number of cycles of the LSTM cell. The LSTM utilizes the previous and

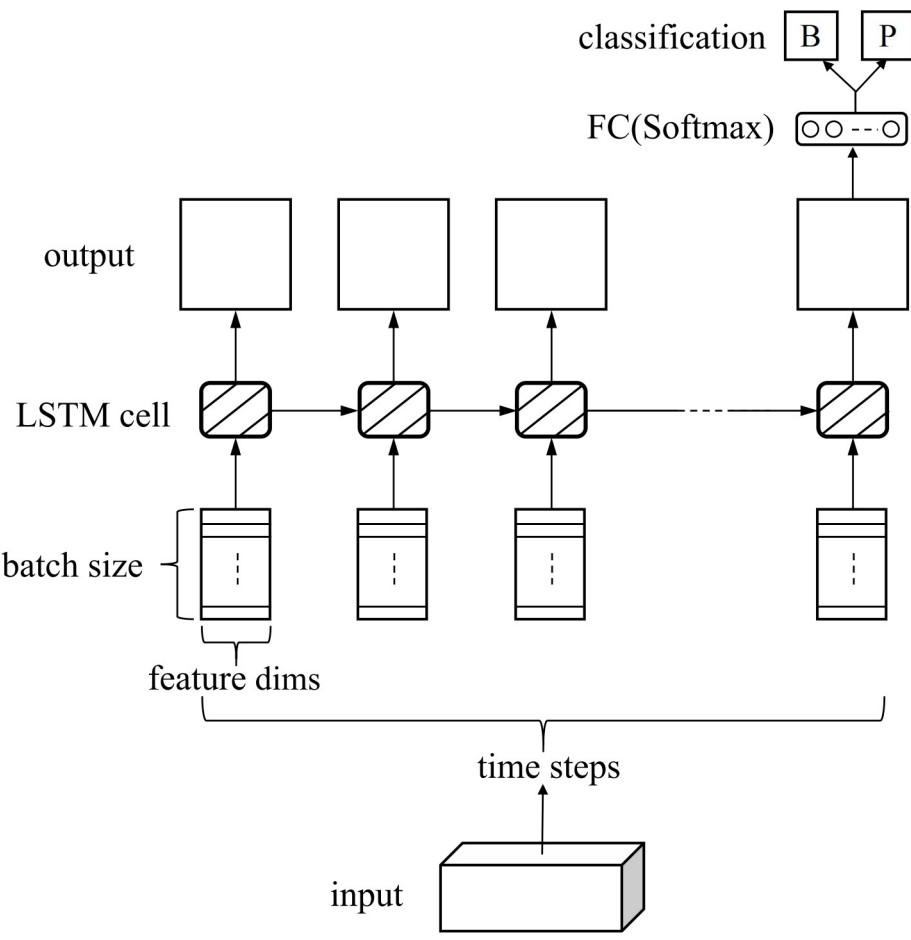

**Fig 7. Structure of the LSTM classifier.**

current information based on the gate structure to generate output vectors. We include a fully-connected layer after the LSTM, which reduces the network's output to 2D. Moreover, we utilize a softmax function to constrain the outputs to sum to 1. Each output corresponds to the confidence of bronchitis and pneumonia.

## Results

### Implementation details

**Datasets.** In the experiments, we adopt six datasets for performance comparison. Table 3 shows the statistics of the datasets. After time shifting, pitch shifting, and noise adding data augmentation, the AR, BR, CR, DR and ER datasets are obtained. The five datasets all include the cough chunks from the RAW dataset. In addition, AR, BR and CR datasets include cough chunks after time shifting, pitch shifting, and noise adding, respectively. DR dataset includes cough chunks from BR and CR datasets. ER dataset includes cough chunks from AR and DR datasets. Each dataset is divided into two non-overlapping groups: training and test data.

**Metrics.** We compare the classification performance on the accuracy, precision, recall, F1-score, and AUC. *Accuracy* is defined as the ratio of the number of samples correctly classified by the classifier to the total number of samples for the given test data. *Precision*, also called positive predictive value, indicates how many of those predicted as bronchitis/pneumonia

**Table 3. Statistics of the six datasets.**

|  | RAW | time shifting | pitch shifting | noise adding | #chunks |
|---|---|---|---|---|---|
| RAW | √ | × | × | × | 410 |
| AR | √ | √ | × | × | 820 |
| BR | √ | × | √ | × | 820 |
| CR | √ | × | × | √ | 820 |
| DR | √ | × | √ | √ | 1230 |
| ER | √ | √ | √ | √ | 1,640 |

patients by the algorithm are bronchitis/pneumonia patients. *Recall* indicates how many of those who are bronchitis/pneumonia patients are predicted by the algorithm. Sometimes, precision and recall are very different, so a comprehensive evaluation metric, a harmonic average of precision and recall, called *F1-score*, is needed. The higher the F1-score, the better the classification performance. Moreover, the *AUC* indicates which model has a higher classified ability.

**Experimented setting.** We take XGBoost [30], RF [31], RNN [33] and GRU [34] as comparison methods. The LSTM, RNN, and GRU networks use the same parameter settings: 32 hidden layers, the learning rate is 0.001, the batch size is 32, the time step is 50, and the Adam optimizer is used to optimize the cross-entropy cost function.

## Accuracy, precision, recall and F1-score test of the predictive model

We first perform experiments on the RAW dataset. As shown in Table 4, we have counted the results of 45 random test experiments and obtained the mean and standard deviation of classification accuracy. In addition to the SVM and LSTM, we compared XGBoost, RF, RNN and GRU to classify. Comparing these classifiers with our model, we see that the classification accuracy is the highest when adopting the SVM classifier, which is 86.04% and the standard deviation is 4.7%. The precision of bronchitis and pneumonia are 93.75% and 87.5%, and recall of them are 88.24% and 93.33%, F1-score are 90.91% and 90.32%. It shows that SVM performs well on disease classification.

As the number of support vectors is small, a tiny scale of training data can train SVM. However, other classifiers have a relatively higher number of parameters and need more samples to update the parameters. Because only support vectors determine the separating hyperplane of the SVM, other samples do not. Moving support vectors will affect the result. However, the

**Table 4. Accuracy, precision, recall and F1-score comparison between SVM and contrast models on the RAW dataset.**

|  | Accuracy(%) | Class | Precision(%) | Recall(%) | F1-score(%) |
|---|---|---|---|---|---|
| SVM | **86.04±4.7** | B | **93.75** | 88.24 | **90.91** |
|  |  | P | 87.5 | **93.33** | 90.32 |
| XGBoost | 76.25±5.26 | B | 82.35 | 82.35 | 82.35 |
|  |  | P | 80 | 80 | 80 |
| RF | 73.54±4.85 | B | 75 | 88.24 | 81.08 |
|  |  | P | 83.33 | 66.67 | 74.07 |
| LSTM | 73.75±3.88 | B | 75 | 88.24 | 81.08 |
|  |  | P | 83.33 | 66.67 | 74.07 |
| RNN | 66.67±5.98 | B | 72.22 | 76.47 | 74.07 |
|  |  | P | 71.43 | 66.67 | 68.97 |
| GRU | 71.46±4.4 | B | 73.68 | 82.35 | 77.77 |
|  |  | P | 76.92 | 66.67 | 71.43 |

result will not be influenced if other samples move beyond the margin boundary. Next, to solve the problem of the small scale of the dataset, we made additional data augmentation.

## Data augmentation

We have adopted data augmentation to increase the number of data to classify bronchitis and pneumonia patients in children. To observe the effect of data augmentation on disease classification, we compared the classification accuracy of six classifiers on six datasets. As shown in Fig 8, in all datasets, the accuracy of SVM is higher than the XGBoost and RF. However, after adding samples, the accuracy of SVM declines since the support vectors are replaced, changing the separating hyperplane.

Besides, compared with the RAW dataset, the accuracy of LSTM is improved on the augmented datasets. The classification accuracy of LSTM on the BR dataset is close to 90%, reaching the best result. Moreover, the classification performance of GRU and LSTM is better than that of RNN. This is because data augmentation increases the number of samples and expands the scale of the dataset. Moreover, GRU and LSTM use gate structures and selectively retain or discard the information.

## Different hyper-parameters

We mainly discuss how different hyper-parameters will impact the accuracy of the CFCS on the RAW dataset. As shown in Fig 9, the value of the regularization parameter $C$ does not affect the result when using a linear kernel. While using the RBF, the accuracy reaches more than 90% when the gamma parameter is $10^{-6}$ and the regularization parameter $C$ is $10^{4}$.

This result indicates that bronchitis and pneumonia have similar characteristics, are difficult to classify in linear feature space, and are easier to classify in high-dimensional feature space. Different gamma parameters can affect the classification accuracy when using RBF, with fixed regularization parameter $C$. When the gamma parameter is $10^{-5}$, the Gaussian distribution in the new feature space is long and thin, and the classification accuracy of unlabeled samples is terrible. The model classification performance is best when the gamma parameter is appropriately reduced to $10^{-6}$. When we continue to reduce the gamma parameter to $10^{-7}$, the Gaussian distribution will be too smooth, affecting the classification performance.

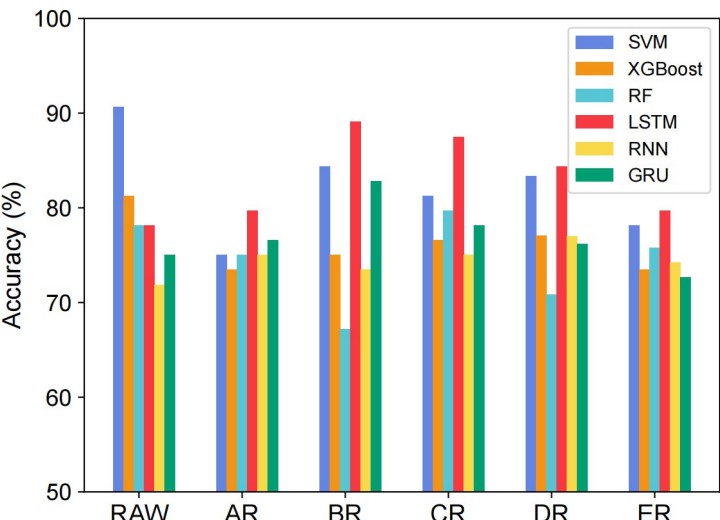

**Fig 8. Accuracy comparison of LSTM and contrast models using different datasets.**

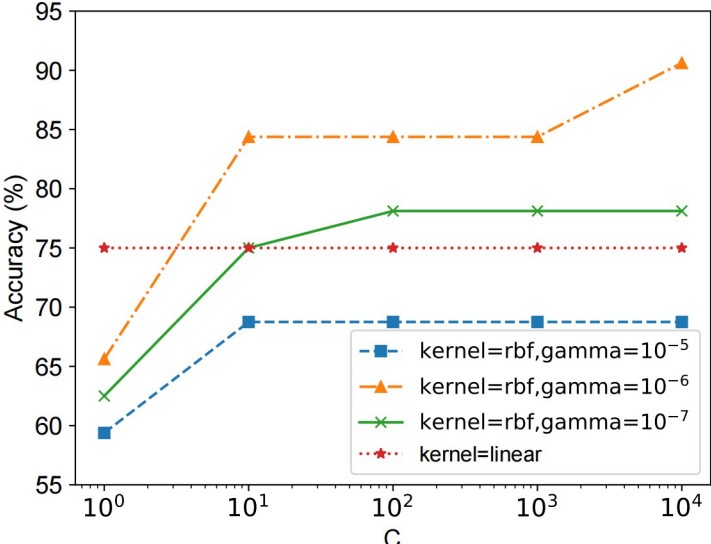

**Fig 9. Accuracy of the SVM model using different hyper-parameters.**

## ROC curves

To further quantify the classification performance of SVM and LSTM under different thresholds, we scan the threshold values of the classifiers and draw the ROC curves on the five augmented datasets. As shown in Fig 10, for both SVM and LSTM classifiers, the AUC on the BR dataset is the highest, 0.92 and 0.93, respectively. The AUC on DR dataset is lower than that on the BR dataset, possibly because of the addition of noise data, which affects classification performance. In addition, it may be that some vital information will be lost after time-shifting, so the AUC on ER dataset is not good as that on the DR dataset.

## MFCC feature representation

Fig 11 shows the MFCC representation: (A) and (B) show the MFCC representations of the two cough chunks in one patient audio, and (C) shows the aggregated feature matrix of the

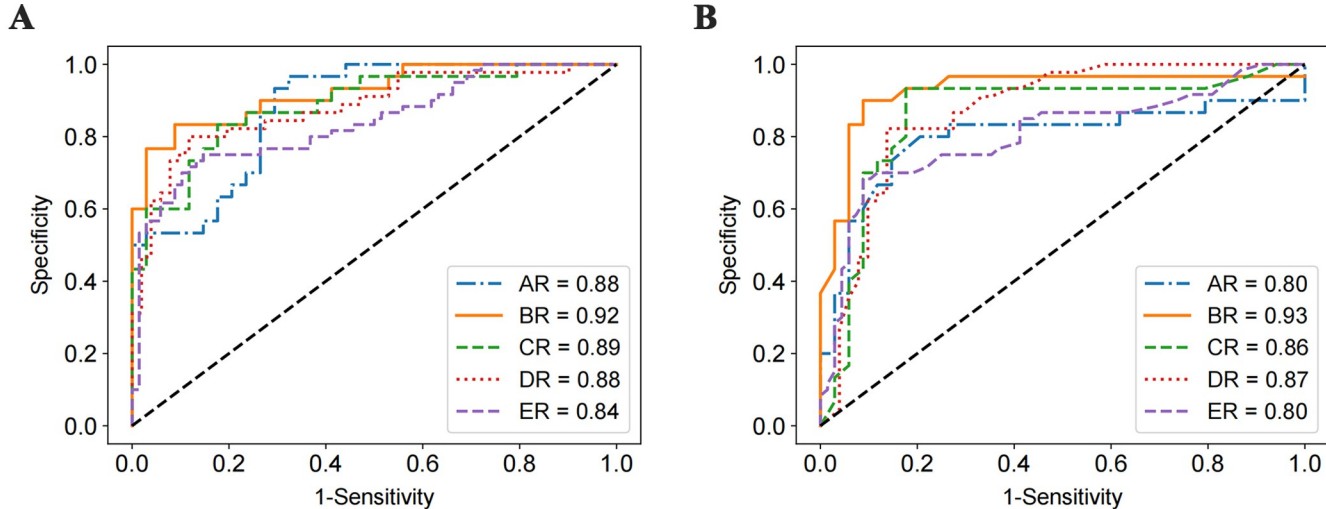

**Fig 10. ROC curves comparison of SVM and LSTM results under different augmentation datasets.** (A) SVM; (B) LSTM.

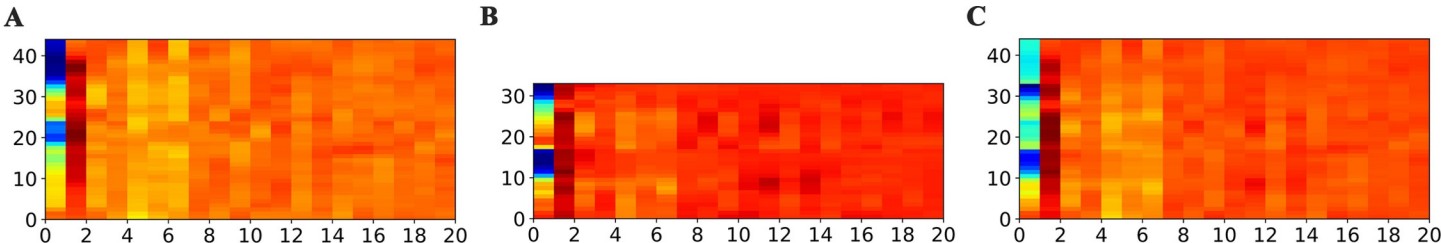

**Fig 11. MFCC features on the heat map.** (A) and (B) show the MFCC representations of the two cough chunks in one patient audio, and (C) shows the aggregated feature matrix of the patient audio. (A) Chunk a; (B) Chunk b; (C) Aggregated features A&B.

patient audio. The experimental results show that it is effective to classify bronchitis and pneumonia using the aggregated results, which represent the features of the patient audios.

## Confusion matrices

Fig 12 is a quantitative comparison of the proposed framework under SVM and LSTM classifiers using the RAW and BR datasets. The RAW test set includes 17 bronchitis and 15 pneumonia patients. From Fig 12A and 12B, both SVM and LSTM misclassify two bronchitis patients as pneumonia. SVM misclassified only one case when classifying pneumonia, while LSTM

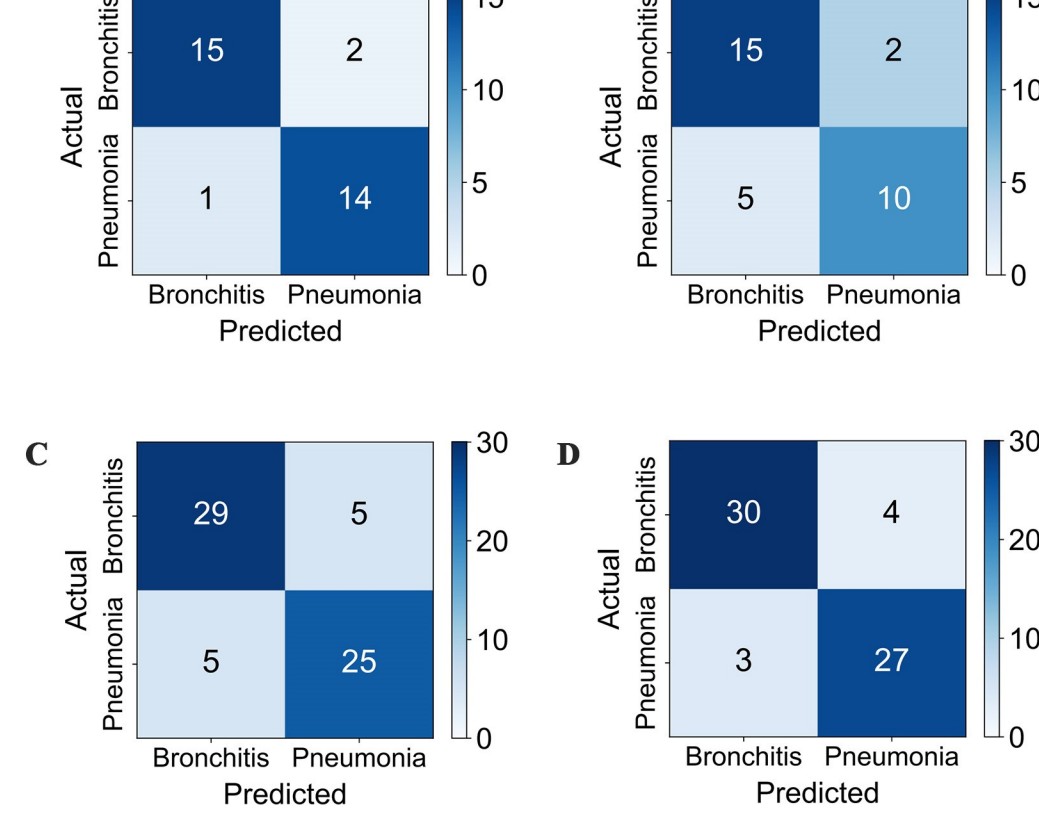

**Fig 12. Confusion matrices of the SVM and LSTM classification results on the RAW and BR datasets.** (A) Results on the RAW dataset: SVM; (B) Results on the RAW dataset: LSTM; (C) Results on the BR dataset: SVM; (D) Results on the BR dataset: LSTM.

misclassified 5 cases. The SVM classifier can achieve a better classification effect in a small dataset. As shown in Fig 12C and 12D, the number of LSTM misclassification in the BR dataset is less than that of SVM misclassification. Data augmentation can increase the number of samples in the dataset, improving the LSTM classifier's classification effect.

## Discussion

In this study, the classification accuracy of SVM on the RAW dataset is 86.04% and the standard deviation is 4.7%. The precision of bronchitis and pneumonia are 93.75% and 87.5%, and recall of bronchitis and pneumonia are 88.24% and 93.33%, which verify CFCS is effective and performs well in the classification of bronchitis and pneumonia patients in children. This result suggests that it is feasible to take the feature of aggregation in each cough chunk as the feature of the diseases. The CFCS can overcome the problem of the influence of useless cough chunks and avoid doctors spending a significant amount of time on labeling. Furthermore, our CFCS can be used as pre-triage to save patients time and improve diagnostic efficiency.

Many researches use SVM, RF, XGBoost and LSTM to classify pneumonia by analyzing cough sounds. Feng K et al. [35] extract features from the audio signal and then use machine learning and deep learning models, like SVM, KNN, and RNN, to diagnose COVID-19 from audio recordings. Rahman D et al. [36] trying several modelling techniques to classify COVID-19 using cough sounds. Compared with XGBoost, SVM can achieve the best result when combined with NMF-Spectrogram feature and undersampling method. Vrindavanam J et al. [37] demonstrate three machine learning classification models SVM, RF and Logistic Regression to identify COVID-19 patients by analyzing cough audio samples. Pahar M et al. [38] use seven machine learning classifiers including SVM and LSTM to discriminate COVID-19 positive coughs from both COVID-19 negative and healthy coughs recorded on a smartphone.

The above paper does not compare LSTM with deep neural networks that process sequence data. Like LSTM, RNN (Recurrent Neural Network) and GRU (Gate Recurrent Unit) are neural networks used to process sequence data. GRU structure is similar to LSTM. As the hidden unit of RNN, GRU has fewer parameters than LSTM and is more difficult to overfit [39].

In order to test the effect of SVM machine learning classification, we also used RF and XGBoost as comparison. In order to test the effect of LSTM deep learning classification, we take GRU and RNN as comparison methods to conduct experiments.

Data augmentation aims to improve the classification effect of the LSTM method. LSTM is a deep learning method that relies on a large-scale dataset. Our experimental results show that the classification accuracy of the LSTM method is indeed improved after data augmentation. However, we can also see that data augmentation has little effect on the improvement of the SVM method. This is because the SVM is not deep learning but a method based on statistical learning. Therefore, we conclude that data augmentation can improve the LSTM method based on deep learning, not the SVM method based on statistical learning.

## Conclusion

In this paper, we propose a novel framework named CFCS for classifying pediatric patients with bronchitis and pneumonia. The proposed framework not only addresses the influence of useless cough chunks but also acquires strong capability in recognizing patient audios. The results of extensive experiments demonstrate the proposed method's high accuracy. We will improve CFCS and apply it to more application scenarios in the upcoming work.

## Author Contributions

**Conceptualization:** Siqi Liao, Chao Song.

**Data curation:** Siqi Liao.

**Formal analysis:** Siqi Liao, Chao Song.

**Funding acquisition:** Chao Song, Yanyun Wang.

**Resources:** Xiaoqin Wang, Yanyun Wang.

**Supervision:** Xiaoqin Wang, Yanyun Wang.

**Writing – original draft:** Siqi Liao.

**Writing – review & editing:** Chao Song, Xiaoqin Wang, Yanyun Wang.

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
