## [Decision Letter · Decision Letter 0]

7 Jul 2022

PONE-D-22-17915A classification framework for identifying bronchitis and pneumonia in children based on a small-scale cough sounds datasetPLOS ONE

Dear Dr. wang,

Thank you for submitting your manuscript to PLOS ONE. After careful consideration, we feel that it has merit but does not fully meet PLOS ONE’s publication criteria as it currently stands. Therefore, we invite you to submit a revised version of the manuscript that addresses the points raised during the review process.

We look forward to receiving your revised manuscript.

Kind regards,

Sathishkumar V E

Academic Editor

PLOS ONE

Journal Requirements:

2. Please include in the Methods section information on whether the IRB approved the verbal consent procedures, and whether these were witnessed or recorded in some way.

Reviewers' comments:

Reviewer's Responses to Questions

**Comments to the Author**

1. Is the manuscript technically sound, and do the data support the conclusions?

Reviewer #1: Partly

Reviewer #2: Partly

2. Has the statistical analysis been performed appropriately and rigorously? 

Reviewer #1: No

Reviewer #2: Yes

3. Have the authors made all data underlying the findings in their manuscript fully available?

Reviewer #1: Yes

Reviewer #2: No

4. Is the manuscript presented in an intelligible fashion and written in standard English?

Reviewer #1: Yes

Reviewer #2: No

5. Review Comments to the Author

Reviewer #1: The authors proposed an audio classification methodology to define respiratory diseases (bronchitis and pneumonia)

Major concerns

In the paper, the audio segmentation is not an agnostic approach; it depends on an empirical analysis.

The authors argue that the LSTM presented better results by increasing the dataset. Although it is a fact, the obtained results were below those reached by the SVM w/o data augmentation. Then, the data augmentation step was not helpful in this case.

It is not possible to make conclusions about the obtained results because the authors did not provide a statistical significance analysis of the results. For instance, it is crucial to present at least the standard deviation or a box plot from the obtained results.

I would like to see comparisons against other state-of-the-art architectures/methods used for audio classification to corroborate the work better. The authors just presented an analysis varying the supervised classifiers.

I also suggest that the authors try to solve a more complex problem. For instance, consider classifying the six classes of the dataset and apply data augmentation only to classes with fewer samples (e.g., Asthmatic bronchitis, Bronchiolitis, Bronchopneumonia, and Lobular pneumonia). This will be a much more interesting analysis.

Minor concerns

It is unclear in the text what AR, BR, CR, DR, and ER stand for. I know that it refers to the data augmentation operation.

Whereas the topic is relevant, the submission lacks novelty for being accepted as a journal contribution.

Reviewer #2: They idea of the paper is interesting and useful. However:

-Other highly cited works on the topic are ignored. Thus, although we acknowledge that they are using a new dataset, don't compare their results with any work in the literature.

-The data Augmentation methods should be further explained and justified.

-The different CNN and RNN alternatives need a much better description.

-English should be revised by a native speaker.

6. PLOS authors have the option to publish the peer review history of their article (what does this mean?). If published, this will include your full peer review and any attached files.

Reviewer #1: No

Reviewer #2: **Yes: **Anton Civit

---

## [Author Response · Author response to Decision Letter 0]

26 Aug 2022

Responds to the editor’s comments:

1. Response to comment: Please ensure that your manuscript meets PLOS ONE's style requirements, including those for file naming.

Response: Thank you for your comment. We have revised the manuscript according to the requirements of PLOS ONE's style, which were remarked in red. 

2. Response to comment: Please include in the Methods section information on whether the IRB approved the verbal consent procedures, and whether these were witnessed or recorded in some way.

Response: Thank you for your comment. We have added the information to the manuscript.

In METHOD section: “the patient audios used in this work are collected from the West China Second University Hospital, Sichuan University, Chengdu, China. The Ethics Committees of West China Second University Hospital approved the study and the verbal consent procedures. Verbal informed consent was obtained from legal guardians of all participants and recorded with the recorder.”

3. Response to comment: Please note that PLOS ONE has specific guidelines on code sharing for submissions in which author-generated code underpins the findings in the manuscript. In these cases, all author-generated code must be made available without restrictions upon publication of the work. Please review our guidelines at https://journals.plos.org/plosone/s/materials-and-software-sharing#loc-sharing-code and ensure that your code is shared in a way that follows best practice and facilitates reproducibility and reuse.

Response: Thank you for your comment. We have uploaded our code to Github according guidelines on code sharing. 

Please see https://github.com/SqLiao/A-bronchitis-and-pneumonia-classification-framework.git.

4. Response to comment: We note that the grant information you provided in the ‘Funding Information’ and ‘Financial Disclosure’ sections do not match.

Response: We are very sorry for our negligence. We will correct it when we resubmit.

5. Response to comment: Please include your full ethics statement in the ‘Methods’ section of your manuscript file. In your statement, please include the full name of the IRB or ethics committee who approved or waived your study, as well as whether or not you obtained informed written or verbal consent. If consent was waived for your study, please include this information in your statement as well.

Response: Thank you for your comment. We have added the information to the manuscript.

In METHOD section: “the patient audios used in this work are collected from the West China Second University Hospital, Sichuan University, Chengdu, China. The Ethics Committees of West China Second University Hospital approved the study and the verbal consent procedures. Verbal informed consent was obtained from legal guardians of all participants and recorded with the recorder.”

Responds to the reviewer’s comments:

Reviewer #1 and Reviewer #2

Response to comment: I would like to see comparisons against other state-of-the-art architectures/methods used for audio classification to corroborate the work better. The authors just presented an analysis varying the supervised classifiers.

Other highly cited works on the topic are ignored. Thus, although we acknowledge that they are using a new dataset, don't compare their results with any work in the literature.

The different CNN and RNN alternatives need a much better description.

Response: Thank you for your professional comment. Many researches use SVM, RF, XGBoost and LSTM to classify pneumonia by analyzing cough sounds. Feng K et al. [1] extract features from the audio signal and then use machine learning and deep learning models, like SVM, KNN, and RNN, to diagnose COVID-19 from audio recordings. Rahman D et al. [2] trying several modelling techniques to classify COVID-19 using cough sounds. Compared with XGBoost, SVM can achieve the best result when combined with NMF-Spectrogram feature and undersampling method. Vrindavanam J et al. [3] demonstrate three machine learning classification models SVM, RF and Logistic Regression to identify COVID-19 patients by analyzing cough audio samples. Pahar M et al. [4] use seven machine learning classifiers including SVM and LSTM to discriminate COVID-19 positive coughs from both COVID-19 negative and healthy coughs recorded on a smartphone.

The above paper does not compare LSTM with deep neural networks that process sequence data. Like LSTM, RNN (Recurrent Neural Network) and GRU (Gate Recurrent Unit) are neural networks used to process sequence data. GRU structure is similar to LSTM. As the hidden unit of RNN, GRU has fewer parameters than LSTM and is more difficult to overfit [5].

In order to test the effect of SVM machine learning classification, we also used RF and XGBoost as comparison. In order to test the effect of LSTM deep learning classification, we take GRU and RNN as comparison methods to conduct experiments.

[1] Feng K, He F, Steinmann J, and Demirkiran I. “Deep-learning Based Approach to Identify Covid-19.” SoutheastCon 2021 (2021): 1-4.

[2] Rahman D and Lestari D. “COVID-19 Classification Using Cough Sounds.” 2021 8th International Conference on Advanced Informatics: Concepts, Theory and Applications (ICAICTA) (2021): 1-6.

[3] Vrindavanam J, Srinath R, Shankar H, and Nagesh G. “Machine Learning based COVID-19 Cough Classification Models - A Comparative Analysis.” 2021 5th International Conference on Computing Methodologies and Communication (ICCMC) (2021): 420-426.

[4] Pahar M, Klopper M, Warren R, and Niesler T. “COVID-19 cough classification using machine learning and global smartphone recordings.” Computers in Biology and Medicine 135 (2021): 104572 - 104572.

[5] Cho, Kyunghyun et al. “Learning Phrase Representations using RNN Encoder–Decoder for Statistical Machine Translation.” EMNLP (2014). 

Reviewer #1

1. Response to comment: In the paper, the audio segmentation is not an agnostic approach; it depends on an empirical analysis.

Response: Thank you for your professional comment. Statistical analysis shows us an energy threshold value to discard speaking voice from cough sounds. Therefore, we used this energy threshold to segment cough sounds. Fig. 5C shows the waveform after patient audios segmentation according to the threshold.

2. Response to comment: The authors argue that the LSTM presented better results by increasing the dataset. Although it is a fact, the obtained results were below those reached by the SVM w/o data augmentation. Then, the data augmentation step was not helpful in this case.

Response: Thank you for your professional comment. The purpose of data augmentation is to improve the classification effect of LSTM method. LSTM is a deep learning method, which relies on a large-scale dataset. From our experimental results, it can show that the classification accuracy of LSTM method is indeed improved after data augmentation. However, we can also see that data augmentation has little effect on the improvement of SVM method. This is because the SVM is not deep learning, but a method based on statistical learning. Therefore, we conclude that data augmentation can improve the LSTM method based on deep learning, but not the SVM method based on statistical learning.

3. Response to comment: It is not possible to make conclusions about the obtained results because the authors did not provide a statistical significance analysis of the results. For instance, it is crucial to present at least the standard deviation or a box plot from the obtained results.

Response: Thank you for your professional comment. We have added the standard deviation from the obtained results. 

4. Response to comment: I also suggest that the authors try to solve a more complex problem. For instance, consider classifying the six classes of the dataset and apply data augmentation only to classes with fewer samples (e.g., Asthmatic bronchitis, Bronchiolitis, Bronchopneumonia, and Lobular pneumonia). This will be a much more interesting analysis.

Response: Thank you for your professional comment. In this paper, we aims to quickly screening two major diseases in children and guiding the next step of diagnosis and treatment, so as to save time for patients. The classification problem of the six classes you suggested is very interesting and we will continue to study it in the following work.

5. Response to comment: It is unclear in the text what AR, BR, CR, DR, and ER stand for. I know that it refers to the data augmentation operation.

Response: Thank you for your professional comment. After time shifting, pitch shifting, and noise adding data augmentation, the AR, BR, CR, DR and ER datasets are obtained. The five datasets all include the cough chunks from RAW dataset. In addition, AR, BR and CR datasets include cough chunks after time shifting, pitch shifting, and noise adding respectively. DR dataset include cough chunks from BR and CR datasets. ER dataset include cough chunks from AR and DR datasets.

6. Response to comment: Whereas the topic is relevant, the submission lacks novelty for being accepted as a journal contribution.

Response: Thank you for your professional comment. It's not easy for us to identify cough sounds diseases, and we've solved two big challenges: (i) the dataset collected from the hospital is small, and (ii) the matching problem of disease labels and cough sounds' features, that is, disease labels do not correspond to features of cough sounds. A small dataset may lead to a low classification accuracy of the diseases. More seriously, mismatch between labels and features may result in incorrect classification results. 

Reviewer #2

1. Response to comment: The data augmentation methods should be further explained and justified.

Response: Thank you for your professional comment. As medical data is a kind of private data, it is expensive to collect such private data. And deep learning relies on a large-scale dataset. Data augmentation aims to increase the number and diversity of training data to improve the robustness of deep learning models.

Time shifting: We adopt time shifting to increase the number of samples of the RAW dataset. This operation can be seen as deleting a small portion of cough sound information to obtain new samples. Time shifting deletes the information at the beginning or end of the cough chunks, where the period ranges from 0 to 0.1s, and fills fixed frequency to keep the duration unchanged.

Pitch shifting: We also adopt pitch shifting to increase the number of samples of the RAW dataset. Raise the pitch of the cough chunks within five half-tones. That is, turn up the frequency. The higher the pitch, the higher the frequency. We can obtain new cough sounds through raw cough chunks by conducting pitch shifting.

Noise adding (white / pink noise): To increase the diversity of samples of the RAW dataset, we mix noise with the original sounds. This operation can be seen as changing the SNR distribution of each cough chunk. Mix each cough chunk with white noise or pink noise. White noise contains various characteristics of noise. Pink noise is the most common noise in nature, and traffic sound can be simulated by pink noise. So we mix the white and pink noise to the cough chunks to obtain new cough sounds.

2. Response to comment: English should be revised by a native speaker.

Response: Thank you for your professional comment. We have carefully checked and improved the English writing in the revised manuscript.

---

## [Decision Letter · Decision Letter 1]

19 Sep 2022

A classification framework for identifying bronchitis and pneumonia in children based on a small-scale cough sounds dataset

PONE-D-22-17915R1

Dear Dr. wang,

We’re pleased to inform you that your manuscript has been judged scientifically suitable for publication and will be formally accepted for publication once it meets all outstanding technical requirements.

Kind regards,

Sathishkumar V E

Academic Editor

PLOS ONE

Additional Editor Comments (optional):

Reviewers' comments:

Reviewer's Responses to Questions

**Comments to the Author**

1. If the authors have adequately addressed your comments raised in a previous round of review and you feel that this manuscript is now acceptable for publication, you may indicate that here to bypass the “Comments to the Author” section, enter your conflict of interest statement in the “Confidential to Editor” section, and submit your "Accept" recommendation.

Reviewer #3: (No Response)

Reviewer #4: (No Response)

2. Is the manuscript technically sound, and do the data support the conclusions?

Reviewer #3: (No Response)

Reviewer #4: (No Response)

3. Has the statistical analysis been performed appropriately and rigorously? 

Reviewer #3: (No Response)

Reviewer #4: (No Response)

4. Have the authors made all data underlying the findings in their manuscript fully available?

Reviewer #3: (No Response)

Reviewer #4: (No Response)

5. Is the manuscript presented in an intelligible fashion and written in standard English?

Reviewer #3: (No Response)

Reviewer #4: (No Response)

6. Review Comments to the Author

Reviewer #3: (No Response)

Reviewer #4: (No Response)

7. PLOS authors have the option to publish the peer review history of their article (what does this mean?). If published, this will include your full peer review and any attached files.

Reviewer #3: **Yes: **Usha Moorthy

Reviewer #4: **Yes: **Indra Raj Upadhyaya

---

## [Editor Report · Acceptance letter]

18 Oct 2022

PONE-D-22-17915R1 

A classification framework for identifying bronchitis and pneumonia in children based on a small-scale cough sounds dataset 

Dear Dr. Wang:

I'm pleased to inform you that your manuscript has been deemed suitable for publication in PLOS ONE. Congratulations! Your manuscript is now with our production department. 

Kind regards, 

on behalf of

Dr. Sathishkumar V E 

Academic Editor

PLOS ONE